

# The role of melatonergic system in intervertebral disc degeneration and its association with low back pain: a clinical study

Chong Chen[1,*], Zongyuan Deng[1,*], Zhengran Yu[1], Yifan Chen[1], Tao Yu[1], Changxiang Liang[1], Yongyu Ye[1], Yongxiong Huang[1], Feng-Juan Lyu[2], Guoyan Liang[1] and Yunbing Chang[1]

[1] Department of Spine Surgery, Guangdong Provincial People's Hospital (Guangdong Academy of Medical Sciences), Southern Medical University, Guangzhou, Guangdong, China
[2] Joint Center for Regenerative Medicine Research of South China University of Technology and The University of Western Australia, School of Medicine, South China University of Technology, Guangzhou, Guangdong, China
[*] These authors contributed equally to this work.

Corresponding authors
Guoyan Liang, liang-guoyan@gdph.org.cn
Yunbing Chang, changyunbing@gdph.org.cn

## ABSTRACT

**Objective**. The mechanisms of intervertebral disc degeneration (IVDD) in low back pain (LBP) patients are multiples. In this study, we attempt to investigate whether melatonergic system plays a potential role in IVDD patients with LBP by analyzing their clinical specimens. The fucus will be given to the correlation between the melatonin receptor expression and intervertebral disc tissue apoptosis.

**Methods**. In this clinical study, 107 lumbar intervertebral disc nucleus pulposus (NP) specimens from patients with LBP were collected with patients' consents. The disc height (DH) discrepancy ratio, range of motion and sagittal parameters of the pathological plane were measured and Pfirrmann grade was used to classified the grades of IVDD level. Discs at grades 1–3 were served as normal control and grades 4–5 were considered as IVDD. The expression levels of melatonin receptor 1A (MT1) and 1B (MT2) were measured by immunohistochemistry. The apoptosis of NP was assessed using TUNEL staining. Their potential associations among MT1/2, DH, apoptosis, sagittal parameters with IVDD and LBP were evaluated with statistical analysis.

**Results**. The incidence of IVDD was positively associated with age and negatively related to VAS scores for LBP ($p < 0.001$). Patients with higher degree of IVDD also have higher DH discrepancy ratio ($p < 0.001$), higher prevalence of lumbar instability ($p = 0.003$) and higher cell apoptosis compared to the control. Nevertheless, no statistically significant correlation was identified between Pfirrmann grade and lumbar sagittal parameters. MT1 and MT2 both were highly expressed in the NP tissues. Importantly, MT1 expression but not MT2 was significantly increased in the intervertebral disc tissue of patients with IVDD and its level correlated well with cell apoptosis level and the severity of IVDD as well as lower VAS scores for LBP.

**Conclusion**. The highly elevated MT1 expression was found in NP tissues of patients with IVDD and LBP compared to the control. This phenomenon probably reflects the compensating response of the body to the pathological alteration of the IVDD and LBP. Therefore, these findings provide the novel information to use selective agonists of MT1 to target IVDD and LBP clinically.

## INTRODUCTION

The prevalence of low back pain (LBP) disorder is estimated to affect approximately 540 million individuals globally, which poses a substantial and intricate health concern (*Knezevic et al., 2021*). The etiology of LBP is multifaceted, while the intervertebral disc degeneration (IVDD) is still one of the predominant underlying factors (*Hartvigsen et al., 2018*). The precise cause of IVDD and its pathogenesis remain elusive, with potential associated factors encompassing age, smoking, diabetes, genetic alterations (*Teraguchi et al., 2017*; *Miller, Schmatz & Schultz, 1988*) and apoptosis (*Yang et al., 2020*). Furthermore, the progression of IVDD may be influenced by mechanisms involving inflammation and pain signaling (*Liu et al., 2021*).

Melatonin (N-acetyl-5-methoxytryptamine) is synthesized in mitochondria, therefore, almost all cells including bone marrows have the capacity to produce melatonin (*Tan et al., 2023*). This molecule plays an important role in the regulation of cellular functions, including antioxidant (*Tan et al., 2015*), anti-inflammatory (*Hardeland, 2018*), anti-degenerative (*Sarlak et al., 2013*) and anti-apoptotic activities (*Yang et al., 2015*) *etc.* *Turgut et al. (2003)*, *Turgut et al. (2006)* first reported the potential association between melatonin and IVDD. They observed that pinealectomy which diminishes the circulating melatonin accelerated process of IVDD in chicken while melatonin treatment improved the condition of IVDD model of rats. Subsequently, the accumulated evidence has shown that melatonin plays significant roles in the IVDD process, including protecting vertebral endplate chondrocytes against apoptosis and calcification (*Zhang et al., 2019*), preventing nucleus pulposus (NP) cells apoptosis induced by oxidative stress (*He et al., 2018*) or by excessive mitophagy (*Chen et al., 2019*) and suppressing NF-$\kappa$B signaling pathway which is associated with the increased proinflammatory cytokines (*Chen et al., 2020*). We previously also demonstrated that melatonin was an important promotor in chondrogenic differentiation of bone marrow mesenchymal stem cells (BMSCs) (*Gao et al., 2014*). Additionally, melatonin effectively hinders the proliferation of NP cells through its receptors MT1 and MT2 mediated action (*Li et al., 2017*). Although the pathogenesis of IVDD remains poorly defined, the result from a controlled clinical trial has shown that the standard therapy combined with melatonin increases the treatment efficacy in LBP (*Kurganova & Danilov, 2015*).

Based on these evidences mentioned above, in the current study, we will examine whether melatonergic system alterations involve in LBP in individuals with IVDD by analyzing their clinical specimens. If so, the fucus will be given to the correlation between the melatonin receptors expression and NP tissue apoptosis since a variety of biological functions of melatonin are mediated by its receptors.

## MATERIALS & METHODS

### Patients and demographic characteristics

This study was approved by the Medical Ethical Committee of Guangdong Provincial People's Hospital (Guangdong Academy of Medical Sciences) (Approval Number: KY2020-612-01) (*Liu et al., 2021*). A total 120 patients for lumbar spine surgery, excluding those with certain conditions or previous surgeries, were enrolled in this study. Finally, 107 patients were completed the study, with their NP specimens were collected. Informed written consent to participate in study was taken from all participants. Data about age, gender, body mass index (BMI), visual analog scale for LBP (LBP-VAS) and leg pain (LP-VAS), diabetes history, smoking history, and radiological imaging were also recorded.

### Patients groups

Patients' Magnetic Resonance Imaging T2 weighted images (MRI T2WI) were collected. The degeneration grade of IVDD was graded from 1 to 5 (*Pfirrmann et al., 2001*). We have adopted the grouping method proposed by *Teraguchi et al. (2014)*, *i.e.,* the grade 4 and 5 discs were classified as degenerative discs, while those with grade 3 and lower were categorized as normal discs.

### Radiograph study

In this study, the methodology published by *Son et al. (2021)* was used to compute the disc height (DH) discrepancy ratio at the pathological level.

Briefly, the range of motion of the pathological plane were examined in upright X-ray photography. Furthermore, parameters including pelvic inclination (PI), pelvic tilt (PT), sacral slope (SS), and lumbar lordosis (LL) were assessed. Assessment of MRI and radiograph data were performed by two examiners who were blinded to group allocation. Each examiner measured the same image twice for all parameters. If there was any disagreement about qualitative parameters between the examiners, a conclusion was reached by consensus discussion. The quantitative parameter was determined as the average result of the two examiners.

### Immunohistochemistry

NP specimens were fixed in 10% formalin and cut into 5 $\mu$m sections. NP sections were dried at 60 °C for 2 h, deparaffinized with xylene and then rehydrated. The sections were washed by PBS and incubated with primary antibodies (1:200) (ab167108 and ab115336; Abcam, Cambridge, UK) at 4 °C overnight. After being washed three times with PBS, the sections were incubated with secondary antibodies (1:200) (ab205719 and ab205718; Abcam, Cambridge, UK) for 50 min at room temperature. Three microscopic images were randomly selected for each sample using a diagnostic scanner (3DHISTECH, Panoramic MIDI). The staining sections were then reviewed and scored as follows by a pathologist with over 10 years of experience: the staining color was scored as no coloring particle (0), light-yellow particle (1), brown-yellow particle (2), and brown particle (3). Cells with<5% staining were scored as negative staining (0); 6–25% staining (1); 26–50% staining (2); 51–75% staining (3); >75% staining (4). The final immunoreaction score (IRS) was defined

as the staining ratio score multiplied by the staining color score (*Creytens, 2019*). And the optical density index (ODI) of representative image were calculated using Image-Pro Plus 6.0 software.

### TUNEL staining

The NP samples were fixed with 4% paraformaldehyde at room temperature for 1 h. After being submerged with 3% $H_2O_2$ and 0.1% Triton X-100 for 10 min respectively, NP samples were washed by phosphate buffer saline (PBS) and stained by *in situ* Cell Death Detection Kit (Roche, Indianapolis, IN, USA) according to the manufacturer's instructions. Subsequently, the cells were exposed to 4′,6diamidino-2-phenylindole (DAPI) stain, followed by the visualization of apoptosis-positive cells using a fluorescence microscope. Apoptosis rate (%) = (number of positive cells / total cells) × 100%.

### Statistical analysis

Normally distributed data are expressed as mean ± standard deviation, and non-normally distributed data are expressed as median (interquartile range). The data were analyzed by Ordinary one-way ANOVA test. Scheffe's *post hoc* analysis was used to determine the significant difference between the groups. In addtion, Pearson's chi-square test, Student's *t* test, Mann–Whitney *U* test and nonparametric Kruskal-Wallis *H* test in different data analyses detailed in tables. All statistical analyses were performed by using the statistical package SPSS 26.0 (SPSS Inc., Chicago, IL). A value of *P* <0.05 was set up to be a statistical significance.

## RESULTS

### Demographic information and health characteristics

First, the Pfirrmann classification was applied to MRI images of human lumbar intervertebral discs, and the respresentative results were showed in Fig. 1. Then, the degrees of the IVDD were classived with the method reported by *Teraguchi et al. (2014)*. The results show that among 107 patients, 59 (55.1%) exhibited degenerative discs, while 48 (44.9%) had normal discs. Those patients with degenerative discs were significantly older as well as had lower LBP-VAS score than patients with normal discs ($p < 0.001$). However, no significant differences were observed in LP-VAS score along with BMI, history of diabetes, and smoking history between the groups ($p > 0.05$) based on the current sample size as shown in Table 1.

### The correlation of Pfirrmann grade and radiograph data

Owing some patients having lack of the completed radiograph data, the DH discrepancy ratio were finally computed in 86 instances, the range of motion was computed in 102 instances, and the lumbar sagittal plane parameters were computed in 106 instances.

The correlation of IVDD and radiograph data were analyzed based on all the data collected mentioned above. The results showed a significant increase in the DH discrepancy ratio correlating to the elevated Pfirrmann grades ($p < 0.001$). *Post hoc* analysis verified the significant differences on these parameters between patients with or without IVDD, *i.e.,* the

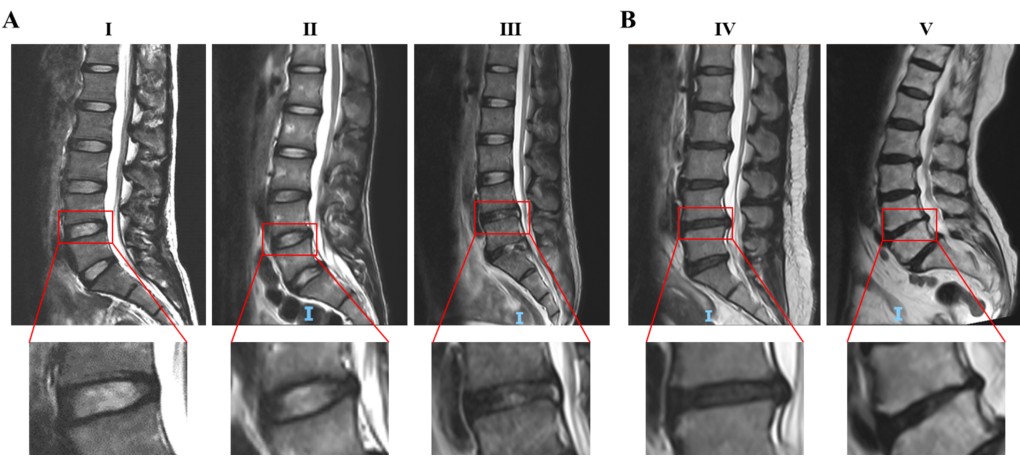

**Figure 1 Representative MRI images of human intervertebral discs of Pfirrmann grade I to V.** (A) Grade III and below were classified as the normal disc. (B) Grade IV and V were classified as the degenerative disc. The red boxes indicate the intervertebral discs for Pfirrmann classification. Classification details of disc degeneration include: structure, distinction of nucleus and anulus, signal intensity, height of intervertebral disc. Abbreviation: MRI, Magnetic Resonance Imaging.

**Table 1 Demographic information and health characteristics between normal and degeneration groups.**

| Characteristics | Total | Normal | Degeneration | P value |
|---|---|---|---|---|
| Patients(n%) | 107 | 48(44.9%) | 59(55.1%) | – |
| Gender | 107 | 48 | 59 | 0.069 |
|    Male | 70 | 36 | 34 | |
|    Female | 37 | 12 | 25 | |
| Age(years) | $43.0 \pm 14.9$ | $36.5 \pm 13.9$ | $48.4 \pm 13.7$ | <0.001[*] |
| BMI | $23.4 \pm 3.4$ | $23.3 \pm 3.8$ | $23.5 \pm 3.1$ | 0.726 |
| Height(m) | $1.66 \pm 0.84$ | $1.67 \pm 0.07$ | $1.66 \pm 0.09$ | 0.376 |
| Weight(kg) | $65 \pm 12.34$ | $65.08 \pm 11.87$ | $64.92 \pm 12.82$ | 0.947 |
| Diabetes | 107 | 48 | 59 | 0.063 |
|    Yes | 5 | 0 | 5 | |
|    No | 102 | 48 | 54 | |
| History of smoking | 107 | 48 | 59 | 0.655 |
|    Yes | 5 | 2 | 3 | |
|    No | 102 | 46 | 56 | |
| VAS, back | 107 | $5.5 \pm 1.1$ | $3.2 \pm 1.4$ | <0.001[*] |
| VAS, leg | 107 | $3.1 \pm 1.1$ | $3.1 \pm 1.3$ | 0.852 |

**Notes.**
Values are expressed as mean $\pm$ SD or $n$.
BMI, body mass index; VAS, Visual analog scale.
Student's t test or Pearson's chi-square test.
*$P < 0.05$.

more severe of the IVDD was, the more prominent of disparity in DH between the supine and standing positions showed. Lumbar instability is defined as lumbar mobility exceeding 10° (*Hanley Jr et al., 1994*). A higher prevalence of lumbar instability was observed in

**Table 2 Analysis of imaging data of patients with different Pfirrmann grades.**

| Pfirrmann grade | II | III | IV | V | P value |
|---|---|---|---|---|---|
| DH discrepancy ratio (%) | $9.81 \pm 6.13$ | $13.17 \pm 3.76$ | $18.37 \pm 4.9$ | $25.61 \pm 2.58$ | $<0.001^{*}$ |
| II($n=5$) | / | $p=0.504$ | $p=0.002$ | $p<0.001$ | |
| III($n=31$) | | / | $p<0.001$ | $p<0.001$ | |
| IV($n=44$) | | | / | $p=0.075$ | |
| V($n=3$) | | | | / | |
| Sagittal plane parameters | | | | | |
| Lumbar lordosis (°) | $27.49 \pm 12.5$ | $30.99 \pm 12.23$ | $31.34 \pm 13.73$ | $30.18 \pm 10.02$ | 0.904 |
| Sacral slope (°) | $26.03 \pm 8.87$ | $28.78 \pm 8.38$ | $29.07 \pm 10.03$ | $28.13 \pm 7.2$ | 0.876 |
| Pelvic tilt (°) | 25.6(10.2) | 24.1(7.6) | 23.7(12.5) | 22.4(6.7) | 0.902 |
| Pelvic incidence (°) | $52.3 \pm 9.77$ | $51.87 \pm 11.83$ | $51.99 \pm 13.82$ | $55.53 \pm 2.54$ | 0.942 |
| Lumbar instability | | Normal | | Degeneration | $0.003^{*}$ |
| Yes ($n=33$) | | 8 | | 25 | |
| No ($n=69$) | | 39 | | 30 | |

Notes.

Values are expressed as mean $\pm$ SD, median (interquartile range) or $n$.

Ordinary one-way ANOVA test followed by Scheffe's post hoc analysis or nonparametric Kruskal-Wallis $H$ test.

$^{*}P < 0.05$.

the patients with IVDD than those without IVDD ($p = 0.003$). Besides, there were no significant differences among different grades in parameters including LL ($p = 0.904$), SS ($p = 0.876$), PT ($p = 0.902$), and PI ($p = 0.942$) between groups. The results were listed in Table 2.

## The association of melatonin receptors expression and IVDD

The potential association of melatonin receptors MT1, MT2 and IVDD was also investigated by using IHC staining. The results showed that both melatonin receptors MT1 and MT2 were present in NP specimens (Fig. 2A). The ODI analysis of the images indicated the variable expression of melatonin receptors among samples correlated to IRS values (Figs. 2B and 2C). MT2 were highly expressed in NP specimens of all individuals without significant differences between degeneration and normal groups ($p = 0.688$). However, MT1 staining score was significantly higher in patients in degeneration group than those in normal group ($p < 0.001$) (Table 3).

## The correlation of melatonin receptors expression and LBP

To investigate the potential modulatory action of melatonergic system on LBP, the correlation of LBP-VAS and LP-VAS score and melatonin receptors expression was analysed. The results were shown in Table 4. It was found that the higher expression level of MT1 and MT2 were associated with lower LBP-VAS and LP-VAS socre in patients ($p < 0.001$). However, there was a lack of substantial correlation between LP-VAS score and differential expression levels of MT1 ($p = 0.11$) or MT2 ($p = 0.114$).

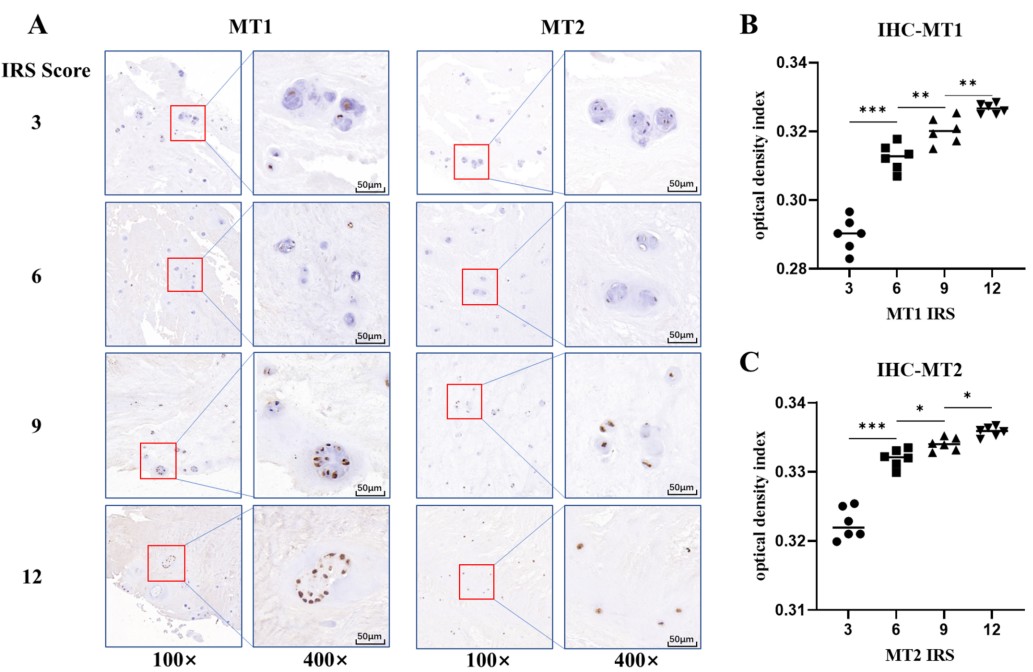

**Figure 2** Immunohistochemistry staining and optical density index analysis of MT1 and MT2. (A) The brown dots represent the staining of MT1 or MT2. The depth of the color was positively associated with its immunoreaction score (IRS) score. (B, C) the optical density index (ODI) analysis of different IRS scores for MT1 and MT2. (* $p$ value $< 0.05$, ** $p$ value $< 0.01$, *** $p$ value $< 0.001$).

## The association of radiographical findings and melatonin receptor expression

A comparison between radiograph data and expression of melatonin receptors was conducted. The results showed that a gradual increase in the DH discrepancy ratio corelated to the increased staining score for MT1/2 ($p < 0.001$). *Post hoc* test revealed highly significant differences between NP tissues with different MT1/2 expression regarding to the DH discrepancy ratio. However, no significant correlation was observed between lumbar instability, sagittal parameters and the expression levels of MT1 and MT2 ($p > 0.05$). It appeared that the expression of MT1/2 in NP tissues are primarily correlated with the DH discrepancy ratio (Table 5).

## Analysis of the association between apoptosis in NP and the Pfirrmann grade

Finally, the association between the apoptosis of NP cells and IVDD was investigated and the results were illustrated in Fig. 3. The cell apoptosis rate of NP cells was determined across various Pfirrmann grades and the results revealed that higher Pfirrmann grade exhibits increased apoptosis rate of NP cells, *i.e.,* significantly elevated apoptosis in NP cells was associated with the severely degenerated intervertebral discs compared to the normal intervertebral discs.

**Table 3  MTNR IRS analysis between normal group and degeneration group.**

| MTNR IRS | 3 | 6 | 9 | 12 | P value |
|---|---|---|---|---|---|
| **MT1** | | | | | |
| Normal (n = 48) | 15(31.3%) | 21(43.8%) | 7(14.6%) | 5(10.4%) | <0.001[*] |
| Degeneration (n = 59) | 7(11.9%) | 14(23.7%) | 23(40.0%) | 15(25.4%) | |
| **MT2** | | | | | |
| Normal (n = 48) | 9(18.8%) | 7(14.6%) | 17(35.4%) | 15(31.3%) | 0.688 |
| Degeneration (n = 59) | 8(13.6%) | 12(20.3%) | 18(30.5%) | 21(35.6%) | |

Notes.
Values are expressed as n (%).
MTNR, melatonin receptor; IRS, immunoreaction score.
Mann–Whitney U test was used between normal and degeneration group.
*P < 0.05.

**Table 4  Visual analog scale analysis between MTNR different IRS.**

| MTNR IRS | 3 | 6 | 9 | 12 | P value |
|---|---|---|---|---|---|
| **MT1** | | | | | |
| VAS, back | 6.2 ± 1.3 | 4.8 ± 0.9 | 3.0 ± 1.1 | 2.9 ± 1.3 | <0.001[*] |
| VAS, leg | 3.6 ± 1.3 | 3.2 ± 1.3 | 2.8 ± 1.0 | 3.1 ± 1.2 | 0.11 |
| **MT2** | | | | | |
| VAS, back | 5.9 ± 1.6 | 4.6 ± 1.4 | 4.2 ± 1.5 | 3.3 ± 1.4 | <0.001[*] |
| VAS, leg | 3.8 ± 1.4 | 3.1 ± 1.0 | 3.0 ± 1.2 | 2.9 ± 1.1 | 0.114 |

Notes.
Values are expressed as mean ± SD.
MTNR, melatonin receptor; IRS, immunoreaction score.
Ordinary one-way ANOVA test was used.
*P < 0.05.

# DISCUSSION AND CONCLUSIONS

To the best of our knowledge, this report probably represents the pioneering attempt to investigate the potential associations among melatonin receptors, IVDD and LBP in patients by utilizing clinical data including radiographic images of spine and NP specimens of patients.

It has been known that the incidence of IVDD is strongly correlated with age. For example, under the age of 50, the prevalence of IVDD was 71% in males and 77% in females but it would surpass 90% in both genders over the age of 50 (*Teraguchi et al., 2014*). Consistent with previous reports, we also observed that patients with IVDD were significantly older than those individuals without IVDD. However, no correlation of IVDD with BMI, diabetes or smoking were detected, respectively.

Some degenerative intervertebral discs can cause discogenic LBP, others are painless depending on the location and severity. In addition, the presence or absence of morphological changes including disc herniation, narrowing, or Schmorl's nodes contribute to the differentiating factors of symptomatic or asymptomatic in patients (*Gopal et al., 2012*). *Son et al. (2021)* proposed that the intervertebral disc height (DH) discrepancy ratio, as determined by standing and supine positions with/without axial loading, could serve as a screening index for discogenic LBP in patients with IVDD. However, they failed

**Table 5  Analysis of imaging data of patients with different MTNR IRS.**

| MTNR IRS | 3 | 6 | 9 | 12 | P value |
|---|---|---|---|---|---|
| **MT1** | | | | | |
| DH discrepancy ratio (%) | $10.96 \pm 2.84$ | $12.38 \pm 3.19$ | $20.30 \pm 3.50$ | $21.60 \pm 1.03$ | $<0.001^{*}$ |
| 3($n = 15$) | / | $p = 0.635$ | $P < 0.001$ | $P < 0.001$ | |
| 6($n = 28$) | | / | $p < 0.001$ | $p < 0.001$ | |
| 9($n = 25$) | | | / | $p = 0.708$ | |
| 12($n = 15$) | | | | / | |
| Lumbar instability | | | | | 0.991 |
| Yes ($n = 33$) | 8(24.2%) | 8(24.2%) | 11(33.3%) | 6(18.2%) | |
| No ($n = 69$) | 12(17.4%) | 26(37.7%) | 18(26.1%) | 13(18.8%) | |
| Sagittal plane parameters | | | | | |
| Lumbar lordosis (°) | $29.52 \pm 13.96$ | $30.83 \pm 11.28$ | $28.93 \pm 12.91$ | $35.40 \pm 13.75$ | 0.334 |
| Sacral slope (°) | $28.10 \pm 10.04$ | $29.06 \pm 8.66$ | $26.35 \pm 8.58$ | $32.30 \pm 9.28$ | 0.155 |
| Pelvic tilt (°) | 25.6(11.7) | 24.2(8.9) | 22.4(12.2) | 22.2(7.9) | 0.424 |
| Pelvic incidence (°) | 56.0(17.2) | 51.2(15.2) | 52.7(16.0) | 56.6(14.4) | 0.591 |
| **MT2** | | | | | |
| DH discrepancy ratio (%) | $9.90 \pm 3.67$ | $14.40 \pm 3.35$ | $16.61 \pm 5.56$ | $19.05 \pm 5.10$ | $<0.001^{*}$ |
| 3($n = 11$) | / | $p = 0.149$ | $p = 0.003$ | $p < 0.001$ | |
| 6($n = 15$) | | / | $p = 0.567$ | $p = 0.034$ | |
| 9($n = 28$) | | | / | $p = 0.314$ | |
| 12($n = 29$) | | | | / | |
| Lumbar instability | | | | | 0.768 |
| Yes ($n = 33$) | 7(21.2%) | 5(15.2%) | 9(27.3%) | 12(36.4%) | |
| No ($n = 69$) | 8(11.6%) | 14(20.3%) | 24(34.8%) | 23(33.3%) | |
| Sagittal plane parameters | | | | | |
| Lumbar lordosis (°) | $29.56 \pm 13.73$ | $32.15 \pm 10.51$ | $31.98 \pm 13.94$ | $29.84 \pm 12.71$ | 0.839 |
| Sacral slope (°) | $28.89 \pm 8.61$ | $29.78 \pm 9.54$ | $29.12 \pm 9.51$ | $27.68 \pm 9.10$ | 0.855 |
| Pelvic tilt (°) | 23.0(9.2) | 18.3(9.2) | 24.2(11.5) | 25.0(8.7) | 0.159 |
| Pelvic incidence (°) | 56.9(20.0) | 52.2(17.2) | 54.1(14.0) | 52.8(15.7) | 0.560 |

**Notes.**

Values are expressed as mean $\pm$ SD, median (interquartile range) or $n$ (%).

MTNR, melatonin receptor; IRS, immunoreaction score; DH, disc height.

Ordinary one-way ANOVA test, Scheffe's post hoc analysis, Mann–Whitney $U$ test or nonparametric Kruskal-Wallis $H$ test was used.

$^{*}P < 0.05$.

to find any significant difference in the degree of IVDD between the LBP (VAS ≥7.0) and the control groups. Nevertheless, other research showed the close relationship between DH and IVDD with disc space narrowing being identified as a characteristic of IVDD (*Dolan & Adams, 2001*). In light with this, we conducted a comparison of the DH discrepancy ratio among patients with varying Pfirrmann grades. Our findings revealed that patients with severe IVDD displayed a higher DH discrepancy ratio and lower LBP-VAS score. This inconsistence may be attributed to the inclusion of non-surgeied individuals with discogenic LBP in the study of *Son et al. (2021)*, as well as the relatively limited sample size. In addition, the specificity or sensitivity of the Pfirrmann grade based on the imaging of MRI sometimes is not sufficient to confirm a diagnosis of discogenic LBP and has some

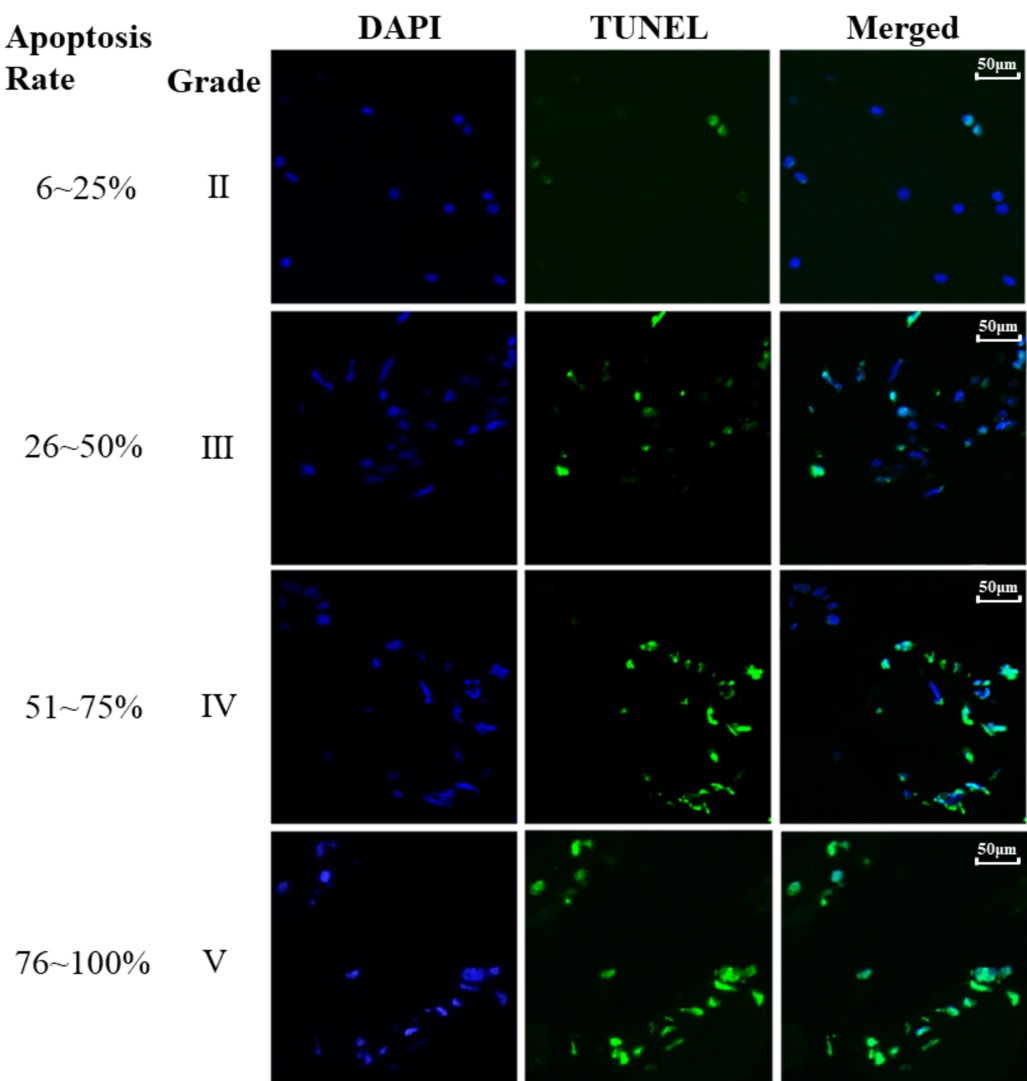

**Figure 3  TUNEL staining of different Pfirrmann grades.** TUNEL assay was performed in NP tissue. The blue staining represent nuclei and the green light represent apoptosis. Apoptosis rate (%) = (number of positive cells / total cells) × 100% was indicated on the left panel of the figure.

shortcomings to fully identify the sources of LBP in those with IVDD (*Ito & Creemers, 2013*). Therefore, the Pfirrmann grades used to classify IVDD can not completely match the clinical symptoms.

Lumbar instability is another parameter to evaluate the severity of IVDD. It indicates the degenerative changes in the intervertebral disc and surrounding structures leading to alterations in the mechanical properties of the functional spinal units (*Galbusera et al., 2014*). It serves as an important risk factor contributing to the occurrence of LBP and often serves as a decision-making index for surgical intervention involving spinal fusion and decompression (*Leone et al., 2007*). We should mention that the debate regarding the clinical definition and diagnostic criteria of lumbar instability remains and it still lacks

universally accepted standard. In this study, we adopted widely accepted criteria of lumbar instability, *i.e.,* sagittal rotation value of 10° and sagittal translation value of 4 mm (*Leone et al., 2007*). Our results indicated a significantly higher prevalence of lumbar instability among individuals with IVDD compared to the control.

Melatonin has been reported to have a protective effect on intervertebral disc cells (*Zhang et al., 2019*; *Chen et al., 2019*), including preventing oxidative stress-induced NP cells apoptosis (*He et al., 2018*), and potentially alleviate chronic pain, particularly LBP (*Danilov & Kurganova, 2016*). Our previous investigation provided evidence of the presence of MT1 and MT2 in the IVDD tissue (*Li et al., 2017*). The involvement of melatonin receptors in pain modulation has been observed in various models of inflammation and neuropathic pain. *Tang et al. (2022)* reported the alleviative effect of melatonin on radicular pain induced by lumbar disc herniation in rats, primarily through its binding to the MT2. The MT2 mediated pain relief activity was also reported in osteoarthritis related pain in rats (*Liu et al., 2022*). Notably, the diminished expression of MT1 weakens the regulatory activity of melatonin on pain perception in rats (*Huang et al., 2013*). In current study, it was found that NP tissues from the patients with IVDD had a significantly upregulated MT1 expression, whereas MT2 expression was not significantly difference between groups. A substantial correlation was observed between increased levels of MT1 and MT2 expression and a significant reduction in LBP-VAS score among patients. Based on the definition of minimum clinically important difference (MCID), a difference exceeding 3.0 points in the LBP-VAS score is considered to have clinical significance (*Ma et al., 2011*). The results showed that the patients with high MT1 staining score from 3 to 12 had the average difference in LBP-VAS score of 3.3 points. In addition, highly elevated NP cells apoptosis found in the severe IVDD suggests that apoptosis plays a role in the progression of IVDD, consistent with previous study (*Miyazaki et al., 2015*). It is well documented that melatonin is a strong antiapoptotic molecule which is at least partially mediated by its MT1/2 receptors (*Radogna et al., 2007*).

Based on all the evidence, we hypothesize that during the initial phases of IVDD, as an adaptive response, NP cells start to upregulate its MT2 expression against this pathological alteration. As IVDD progression, the resultant LBP as a chronic stressor stimulates the expression of MT1. The elevated MT1 level can maximize the analgesic effect of melatonin on LBP as well as inhibits apoptosis in NP cells while the expression of MT2 remains relatively stable throughout this progression. This may be the reason that highly expressed MT2 was observed in all patients without significant differences between groups while significantly high level of MT1 expression was only detected in patients with severe IVDD compared to the control patients. Previous studies have revealed the ubiquitous presence of both MT1 and MT2 across various anatomical regions in the human body, including the brain, retina and other internal organs (*Ahmad et al., 2023*), indicating the important physiopathological roles of melatonergic system in human body. In addition, the three-dimensional structures of the human melatonin receptors MT1 and MT2 unveiled that many melatonin structure-like compounds exhibited a specific affinity towards the MT1 rather than MT2 (*Stauch et al., 2019*; *Johansson et al., 2019*). This characteristic of MT1 implays its relatively high efficiency to modulate LBP in patients with IVDD compared to

the MT2. Our discovery provides better opportunity to develop the novel MT1 agonists to protect against IVDD and its associated LBP with less adverse effects.

In conclusion, in this clinical study, we have identified the highly elevated MT1 expression in NP tissues of patients with IVDD and LBP compared to the control patients, which may be a compensiting response of the body to the pathological alteration of the IVDD and LBP. These findings offer novel perspectives on the therapeutic potential to use MT1 specific agonists to treat IVDD and LBP.

## LIMITATIONS

The study had limitations common to clinical research, being a single-center observational study, so our findings should be confirmed by others. Healthy human intervertebral disc samples were difficult to obtain, so we used *Teraguchi et al.'s (2014)* criteria to classify discs with Pfirrmann grade 3 or lower as the control group. The subjective nature of the Pfirrmann grade system may introduce bias in assessing IVDD, but no better methodology currently exists. This issue should be addressed in the future.

### Funding

This work was supported by the National Natural Science Foundation of China (82102636, 82272552), the Natural Science Foundation of Guangdong Province (2022A1515012557, 2023A1515030001, 2023A1515010403), the Medical Scientific Research Foundation of Guangdong Province (A2023009), and the Guangzhou Municipal Science and Technology Project (202103000053, 2023A04J0500, 2024A04J3697, 2024A04J10010). The funders had no role in study design, data collection and analysis, decision to publish, or preparation of the manuscript.

### Grant Disclosures

The following grant information was disclosed by the authors:
The National Natural Science Foundation of China: 82102636, 82272552.
Natural Science Foundation of Guangdong Province: 2022A1515012557, 2023A1515030001, 2023A1515010403.
Medical Scientific Research Foundation of Guangdong Province: A2023009.
Guangzhou Municipal Science and Technology Project: 202103000053, 2023A04J0500, 2024A04J3697, 2024A04J10010.

### Competing Interests

The authors declare there are no competing interests.

### Author Contributions

- Chong Chen conceived and designed the experiments, prepared figures and/or tables, authored or reviewed drafts of the article, and approved the final draft.

- Zongyuan Deng performed the experiments, analyzed the data, prepared figures and/or tables, and approved the final draft.
- Zhengran Yu performed the experiments, authored or reviewed drafts of the article, and approved the final draft.
- Yifan Chen performed the experiments, prepared figures and/or tables, and approved the final draft.
- Tao Yu analyzed the data, authored or reviewed drafts of the article, and approved the final draft.
- Changxiang Liang analyzed the data, authored or reviewed drafts of the article, and approved the final draft.
- Yongyu Ye performed the experiments, authored or reviewed drafts of the article, and approved the final draft.
- Yongxiong Huang analyzed the data, prepared figures and/or tables, and approved the final draft.
- Feng-Juan Lyu conceived and designed the experiments, authored or reviewed drafts of the article, and approved the final draft.
- Guoyan Liang conceived and designed the experiments, authored or reviewed drafts of the article, and approved the final draft.
- Yunbing Chang conceived and designed the experiments, authored or reviewed drafts of the article, and approved the final draft.

## Human Ethics

The following information was supplied relating to ethical approvals (*i.e.*, approving body and any reference numbers):

The Medical Ethical Committee of Guangdong Provincial People's Hospital (Guangdong Academy of Medical Sciences) granted Ethical approval to carry out the study within its facilities (KY2020-612-01).

## Ethics

The following information was supplied relating to ethical approvals (i.e., approving body and any reference numbers):

The Medical Ethical Committee of Guangdong Provincial People's Hospital

## Data Availability

The raw data are available in the Supplemental Files and at Figshare:

Zongyuan Deng (2024). Raw data.xlsx. figshare. Dataset. https://doi.org/10.6084/m9.figshare.25599972.v1.

## Supplemental Information

Supplemental information for this article can be found online at http://dx.doi.org/10.7717/peerj.17464#supplemental-information.

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
