# Peer review of "The role of melatonergic system in intervertebral disc degeneration and its association with low back pain: a clinical study"

_PeerJ, doi:10.7717/peerj.17464_

## Round 0.1 · original submission · Major Revisions

Dear Authors, There are many major comments from all three peer reviewers that need to be resolved. Please read the comments carefully and revise accordingly and meticulously. Thank You.

**Language Note:** PeerJ staff have identified that the English language needs to be improved. When you prepare your next revision, please either (i) have a colleague who is proficient in English and familiar with the subject matter review your manuscript, or (ii) contact a professional editing service to review your manuscript. PeerJ can provide language editing services - you can contact us at [email protected] for pricing (be sure to provide your manuscript number and title). – PeerJ Staff

Reviewer 1 ·

Basic reporting

1. The title corresponds to the work; however, the title can be reworded to be more appropriate and precise: ‘The Role of the Melatonergic System in Intervertebral Disc Degeneration and its Association with Low Back Pain: A Clinical Study’

2. The abstract summarizes the main findings of the article: however, need to check grammar (singular/plural) terms ; line 28.

3. There are no consistency in terminology throughout the text. For example, the author uses both ‘IVDD’ and intervertebral disc degeneration" interchangeably. Choose one term and stick with it for clarity.

Experimental design

1. The research gap is not focusing on the apoptosis mechanism only, but the author is able to state objectives or aims of the study.
2. The material and methods give quite enough details, but the presentation of methods could be improvised.
3. The results are presented in an acceptable manner but for correlation, the author should specify the statistical methods used to establish these correlations and consider providing correlations coefficients to quantify strength and direction of relationships.
4. There is an undefined abbreviation in line 94 ; MRI T2W1, please advised.

Validity of the findings

1. A sample size of 107 is generally considered reasonably large for many statistical analyses. However, the author is able to show enough comparison of data for assessing intervertebral disc degeneration (IVDD), associated with LBP and melatonin receptor to associate with apoptosis mechanism.
2. Results of melatonin receptors staining and IVDD radiological finding score, author followed well to use mean and standard deviation (SD) for normally distributed data and mean and in the context of medical research, a sample size of 107 is often considered robust.
But the results of the presentation need to be improvised.
i. The figures are not clear, especially for the histology figure; it was incorrectly presented (small size of images)
ii. The dapi staining is not well shown in the images of apoptosis grade 2, should be more nucleus staining in lesser grade.
iii. iii) author is advised to edit the image : adjust or do corrections on the background to a brighter/clearer image
iv. the target /cell of interest, the fluorescence is not striking; the target of interest, because its green fluorescence is visually striking.
v. and there is no figure legend, information about the box, the nucleus staining…
vi. The description of figures was not in order, later numbering (iv & v) was described first rather than describing no iii in Figure 1
vii. No credit given to owner of images/ - or /credit attribution to patient in which hospital taken from…..
viii. some of the tables can be improvised for clearer understanding, author is advised to tabulate side by side for MT1IRS and MT2IRS in table 5 for easier comparison.

Additional comments

COMMENTS ON DISCUSSION
The discussion interprets the findings in view of the results obtained in this and in past studies on this topic and mechanism of melatonergic to induce apoptosis.
The conclusions are valid and based on the results of the study. This limitation is reasonable and common in clinical studies, especially when transitioning from clinical observations to molecular mechanisms. So, it will indicate that while your clinical study provides valuable insights into the association between melatonergic system markers, IVDD, and low back pain, there is a need for further research to associate it into the detailed molecular mechanisms. Including such a limitation is beneficial as it acknowledges the scope of the current study and sets the stage for future investigations.
GENERAL COMMENTS
1. The citing references are adequate but need to follow the format of PEER J
2. Grammar and writing:
The English language should be improved to ensure that an international audience can clearly understand the text.
The grammar need to be refine throughout the whole writing and there are lots of spelling mistakes e.g. ‘heigh’ line 30, ‘reflets’ line 46, line 104, ‘Pfirrmann’, line 303 …..and others in the main text (quite a lot to be listed), there are also unnecessary article used eg. …a grade 1-5 (line 94), a in line 92, , I strongly suggest author to send the improvised writing to English expertise to check the grammar and writing.

There is also unformatted citing writing e.g in line 100 Teraguchi et al (year?)

Reviewer 2 ·

Basic reporting

This is a well written research paper. Professional English has been used throughout . Authors have used diagrams, tables and figures appropriately in the text. They have reviewed relevant literature and have appropriate  references. However, There are still some grammatical errors which need to be improved. Some of these errors (with possible corrections) are: 
Line 65: almost all cells has the capacity to produce melatonin including bone marrows [6] ->almost all cells  including bone marrows have the capacity to produce melatonin [6]
Line 93: "All patients have filled the consent forms."-> informed written consent to participate in study was taken from all participants.
line 105: what does compuse mean. need to used proper word
Line 155: First, the representative MRI images of human discs with different grades of degeneration are collected and the results were shown in Fig->First, the representative MRI images of human discs with different grades of degeneration were collected and the results are shown in Fig
Line 193,194: Conclusion should not be mentioned in results
Line 215: imaging of patients radiograph-> Radiographic images of spine
Line 220: Based on the current data, we have not detected a correlation between IVDD with those factors of BMI, diabetes or smoking, respectively.-> In our study, we didn't find any correlation between IVDD with  BMI, diabetes or smoking.
Line 234,235: Need modification to make clear what authors want to state
Line 272:  IVD tissue starts to upregulate its MT2 expression to against this pathological. -> Either use  to or against.
Line 302:  "come features" -> Inappropriate word
Line 303: "Even" we restrictly followed the study. -> Although we restrictly followed the study,

Experimental design

Research question is well defined, relevant & meaningful. Methods have been described with sufficient information.

Validity of the findings

Al underlying data have been provided; they are robust, statistically sound. Authors have included limitations of their study as well. Conclusions are well stated, linked to original research question.

Additional comments

Consent form is not in English, so we are unable to comment on it.

Reviewer 3 ·

Basic reporting

This manuscript lacks explicit rationales and related hypotheses. Please refer to my additional comments.

Experimental design

Statistical analysis is limited. Please refer to my additional comments.

Validity of the findings

Since detailed statistical analysis lacks, the results/conclusions are compromised. Please refer to my additional comments.

Additional comments

1. Please provide explicit rationales and related hypotheses.
2. How is the sample size decided? How many individuals were recruited in total before exclusion?
3. The authors mention “Owing to the unavailability of specific radiograph data in some patients”, do these missing data influence the results? How to validate?
4. Substantial information regarding detailed statistical analysis is lacking in reporting the results.
5. In Table 3 and Figure 2, please provide more explanations regarding the statistical analysis. Does this table include all scores? The results in “The Association of Melatonin Receptor Expression and IVDD” are not in agreement with the Figure. Please reconfirm.
6. In Line 198-199, what statistical methods are used to test?
7. In Line 207, how is the statistical analysis conducted?
8. In Line 221, it is described “we have not detected a correlation between IVDD with those factors of BMI, diabetes or smoking, respectively”. Does the sample size matter? What would be observed with an increased sample size? Since the sample size is compromised, it is weak to draw the related conclusion.
9. The section of Discussion is redundant, please make it brief.
10. There are too many typos, please correct them.

---

## Round 0.2 · accepted · Accept

Article is now accepted thanks

Reviewer 2 ·

Basic reporting

1. BASIC REPORTING

-Language and Clarity
- The English language used throughout the article is clear and professional.
- The introduction provides relevant context for the study.
- References to relevant literature are appropriately cited.
- The overall structure adheres to journal standards.

Figures and Raw Data
- The figures are relevant, well-labeled, and adequately described.
- Raw data has been provided, as per the journal's policy.

Experimental design

2. EXPERIMENTAL DESIGN

Research Question
- The research question is well-defined and meaningful.
- The study effectively fills an identified knowledge gap.

Rigorous Investigation
- The investigation follows high technical and ethical standards.
- Sufficient detail is provided in the methods section for replication.

Validity of the findings

3. VALIDITY OF THE FINDINGS

Impact and Novelty
- This is in a way a basic science research but it can be a possible starting point of a new group of therapies down the road

Data Robustness
- All underlying data have been provided.
- The data are robust, statistically sound, and well-controlled.

Conclusions
- The conclusions are well-stated, linked to the original research question, and limited to supporting results.

Additional comments

4. General Comments

- The study's focus on melatonin receptors and their potential role in IVDD and LBP is commendable.
- The hypothesis regarding MT1 and MT2 expression in response to IVDD progression is intriguing.

Reviewer 3 ·

Basic reporting

Please refer to my Additional comments

Experimental design

Please refer to my Additional comments

Validity of the findings

Please refer to my Additional comments

Additional comments

The authors has answered my questions briefly.
Thanks.